# Long Term Impact of Epicardial Left Atrial Appendage Ligation on Systemic Hemostasis: *LAA HOMEOSTASIS-2*

**DOI:** 10.3390/jcm11061495

**Published:** 2022-03-09

**Authors:** Krzysztof Bartus, Sri Harsha Kanuri, Radoslaw Litwinowicz, Mehmet Ali Elbey, Joanna Natorska, Michal Zabczyk, Magdalena Bartus, Boguslaw Kapelak, Rakesh Gopinnathannair, Jalaj Garg, Mohit K. Turagam, Maciej T. Malecki, Randall J. Lee, Dhanunjaya Lakkireddy

**Affiliations:** 1Department of Cardiovascular Surgery and Transplantology, Jagiellonian University Medical College, John Paul Hospital, ul. Pradnicka 80, 31-202 Krakow, Poland; krzysztofbartus@gmail.com (K.B.); bogus.kapelak@gmail.com (B.K.); 2Institute of Cardiology, Jagiellonian University Medical College, 31-008 Krakow, Poland; j.natorska@szpitaljp2.krakow.pl (J.N.); michalzabczyk@o2.pl (M.Z.); 3The Kansas City Heart Rhythm Institute and Research Foundation, Overland Park, KS 66211, USA; harsha9009@gmail.com (S.H.K.); malielbey@gmail.com (M.A.E.); drrakeshg@yahoo.com (R.G.); dlakkireddy@gmail.com (D.L.); 4Department of Pharmacology, Jagiellonian University Medical College, 31-531 Krakow, Poland; krolik.mb@gmail.com; 5Division of Cardiovascular Medicine, Medical College of Wisconsin, Milwakee, WI 53226, USA; garg.jalaj@yahoo.com; 6Department of Cardiovascular Medicine, Icahn School of Medicine at Mount Sinai, New York, NY 10029, USA; mohit.turagam@mountsinai.org; 7Department of Metabolic Diseases, Jagiellonian University Medical College, 30-688 Krakow, Poland; maciej.malecki@uj.edu.pl; 8Electrophysiology Section, Department of Cardiology, University of California San Francisco, San Francisco, CA 94143, USA; lee@medicine.ucsf.edu

**Keywords:** left atrial appendage closure (LAAC), LARIAT, blood pressure (BP), neurohormonal changes, systemic hemostasis

## Abstract

Background: Recent data suggest that epicardial left atrial appendage closure (LAAC) is associated with several short-term neurohormonal effects. However, the long-term effects are currently unknown. Objective: To investigate the effects of percutaneous epicardial left atial appendage (LAA) exclusion using LARIAT on neurohormonal profiles at long-term follow-up. Methods: In a prospective single centre study, 60 patients with long-standing, persistent atrial fibrillation (AF) LARIAT were treated. The major hormones of the adrenergic system, renin-angiotensin-aldosterone system (RAAS), and natriuretic peptides were assessed before the intervention and at regular intervals during the following two years. Results: In patients with epicardial LAAC, atrial natriuretic peptide (ANP) levels were significantly increased from baseline at 24 h and decreased at 7 days, 1 month, and 3 months, while remaining unchanged at 12 and 24 months. Noradrenaline levels were significantly lower at 24 h, 7 days, 1 month, 6 months, 12 months, and 24 months, while epinephrine levels decreased significantly at 1 month, 6 months, 12 months, and 24 months. Plasma renin activity significantly decreased at 7 days, 1 month, 6 months, 12 months, and 24 months, while aldosterone levels significantly decreased at 6 months, 12 months, and 24 months. Endothelin-1 and vasopressin showed a significant increase and decrease, respectively, at 24 h, 7 days, 1 month, 6 months, 12 months, and 24 months. There was also a significant decrease in systolic and diastolic blood pressure at 3 months, 6 months, 1 year, and 2 years after the intervention. Conclusions: Epicardial LAAC in AF patients is associated with persistent neurohormonal changes favouring blood pressure reduction.

## 1. Introduction

Atrial fibrillation (AF) is the most common cardiac arrhythmia with an increased predisposition to systemic embolism, such as transient ischaemic attack (TIA) or stroke. In patients with non-valvular AF and a contraindication to oral anti-coagulation (OAC) therapy, epicardial left atrial appendage closure (LAAC) with the LARIAT device (Sentreheart, Redwood City, CA, USA/currently Atricure Inc.) has been used to exclude the left atrial appendage (LAA) and prevent thrombus formation [1]. As a consequence of epicardial LAA exclusion, our group observed several significant differences in the haemodynamics and neurohormonal effects of epicardial LAAC and a significant reduction in systolic blood pressure during 3-month follow-up [2]. With increased application of percutaneous LAAC therapies for stroke prophylaxis, it is currently unknown whether elimination of atrial natriuretic peptide (ANP) secretion from the LAA may have adverse effects on long-term physiological regulation of fluid volume, brain natriuretic peptide (BNP), salt-water balance, renin-angiotensin-aldosterone system (RAAS), adrenergic system, and blood pressure (BP). Therefore, we aimed to investigate the serial changes in the adrenergic system, electrolytes, ANP, BNP, and RAAS during acute, intermediate, and long-term follow-up in patients undergoing percutaneous epicardial LAA ligation with the LARIAT device (Sentreheart, Redwood City, CA, USA/currently Atricure Inc.).

## 2. Methods

### 2.1. Patient Population

This is a single-center prospective study including 60 patients with long-standing persistent AF who underwent percutaneous epicardial LAA ligation with the LARIAT device. Informed consent was obtained from all patients. The protocol was conducted with the approval of the local ethics committee at Jagiellonian University, Krakow, Poland.

### 2.2. Inclusion/Exclusion Criteria

The following criteria were met to be included in this study: (1) age 18 years or older; (2) non-valvular long-standing persistent AF; (3) at least one risk factor for embolic stroke (congestive heart failure, hypertension, age > 65 years, diabetes mellitus, previous TIA, or prior vascular disease); (4) considered poor candidates for long-term anticoagulant therapy; and (5) transthoracic echocardiogram performed within 30 days and 1 year after the procedure.

Patients were excluded from the study if they met the following exclusion criteria: (1) history of cardiac surgery; (2) severe pectus excavatum; (3) myocardial infarction within 3 months; (4) previous embolic event within the last 30 days; (5) New York Heart functional class IV heart failure symptoms; (6) history of chest radiation; (7) atrial–septal defect; (8) patent foramen ovale with atrial septal aneurysm; or (9) mechanical prosthetic heart valve. 

All patients underwent screening contrast computed tomography (CT) to assess the left atria (LA) size and LAA geometry. Based on the CT scan, the following additional exclusion criteria were established: (1) LAA width > 50 mm; (2) a superiorly oriented LAA with LAA apex directed behind the pulmonary trunk; (3) bi-lobed LAA or multi-lobed LAA, in which lobes were oriented in different planes exceeding 50 mm; and (4) a posteriorly rotated heart.

### 2.3. Procedure Details

All 60 patients underwent percutaneous epicardial LAA ligation with the LARIAT device as previously described [3,4]. A transesophageal echocardiogram (TEE) was performed before the procedure to rule out LAA thrombus. 

Blood samples were collected from an antecubital vein before the LARIAT procedure and 24 h, 7 days, 1 month, 3 months, 6 months, 12 months, and 24 months after the procedure. The pre-procedure blood samples were collected after overnight fasting in all patients. The blood samples were collected in vacutainer tubes (tubes were anticoagulated with K3-EDTA for complete blood count, 0.109 M sodium citrate was added for haemostasis and fibrinolysis tests, and serum vacuum tubes were used for routine clinical chemistry laboratory tests and ELISA tests. In addition, anti-factor Xa activity (IU/mL) was measured in all patients before the procedure. All patients were instructed to maintain their usual diet and not to consume any additional salts or carbohydrates for up to 48 h before the blood draw.

The following blood samples were collected: International Normalized Ratio (INR), activated partial thromboplastin time (APTT), high sensitivity C-reactive protein (CRP), sodium, potassium, chloride, and glycated hemoglobin (HbA1c). In addition, NT-proANP (Invitrogen ThermoFisher, Waltham, MA USA), NT-proBNP (Biomedica, Vienna, Austria), adrenaline (IBLInternational, Hamburg, Germany), noradrenaline (IBLInternational, Hamburg, Germany), aldosterone (Cayman, Ann Arbor, MI, USA), plasma renin activity (IBLInternational, Hamburg, Germany), vasopressin (FineTest, Wuhan, China), and endothelin-1 (Cayman)were determined by ELISA.

### 2.4. Clinical Follow Up

Patients were on stable medical BP reduction therapy prior to the procedure and were instructed not to change their medications throughout the follow-up period. All blood pressure medications were routinely continued, with the exception of diuretics, which were discontinued on the day of the procedure. Patient adherence to medication was strictly monitored by interview and patient diary. There were no changes in blood pressure medication that were noted during follow-up that could affect the obtained results.

Blood pressure and heart rate were measured at baseline, 3 months, 6 months, 12 months, and 24 months after the intervention.. All recordings were obtained in the supine position with an automatic blood pressure monitor using the standard BP measurement protocol to avoid discrepancies. 

### 2.5. Statistical Analysis

Continuous variables were presented as mean ± standard deviation or median (interquartile range) while categorical variables were presented as a number or percentages. Continuous variables were compared using Student’s *t*-test or Mann–Whitney U Test. For repeated measures, the ANOVA or Friedman test were used. The categorical variables were compared using the chi-Square test or Fisher’s Exact test. Statistical analysis was performed using IBM SPSS Statistics version 24 (IBM, Armonk, NY, USA). A *p*-value of <0.05 was considered significant. 

## 3. Results

### 3.1. Baseline Characteristics

A total of 60 patients were followed up prospectively over 24 months. The mean age of the patient population was 67.5 ± 8.1 years and 66.7% were men. The body mass index of the patient population was 28 ± 2.9 kg/m^2^. The most common cardiovascular risk factors were hypertension (80%), followed by diabetes (25%), coronary artery disease (CAD) (18.3%), and congestive heart failure (13.3%). The mean CHA_2_DS_2_-VASc score was 4 ± 1.7 and the HAS-BLED score was 3.6 ± 1.2 (Table 1).

### 3.2. Impact of Epicardial LAA Ligation on Electrolytes

The mean baseline values for sodium, potassium, and chloride and the values after 24 h, 7 days, 1 month, 3 months, 6 months, 12 months, and 24 months are shown in Table 2. Compared to baseline, there was a significant decrease in sodium levels at 24 h (138.4 ± 2.6 mmol/L; *p* value < 0.05). However, there was no significant difference in sodium at subsequent time intervals. There were also no significant changes in potassium and chloride levels at any time as compared to baseline (Table 2).

### 3.3. Impact of Epicardia; LAA Closure on Natriuretic Peptides and Adrenaline

NT-proANP: The mean baseline value of NT-proANP measured before the intervention was 9.94 (4.09–16.87) ng/mL. Compared with baseline, there was a significant increase in NT-proANP levels when measured 24 h after the procedure (15.39 (10.93–16.10); *p* value < 0.05). However, there was a significant decrease in the levels of NT-proANP measured at 7 days (6.31 (3.71–11.79); *p* value < 0.05), 1 month (5.44 (4.29–10.58); *p* value < 0.05), and 3 months (4.58 (4.00–5.48); *p* value < 0.05). In addition, there was no significant difference in the levels of NT-proANP measured at 6 months (8.36 (5.10–14.80); *p* value > 0.05), 12 months (9.03 (4.73–13.98); *p* value > 0.05), and 24 months (11.01 (4.98–19.81); *p* value > 0.05) (Table 3 and Table 4). For NT-proANP, there were significant differences in repeated measurements (*p* value < 0.05). 

NT-proBNP: The mean baseline value of NT-proBNP measured before the intervention was 167.8 (63.7–238.7) pg/mL. Compared with baseline, there was no significant difference in the levels of NT-proBNP measured at 24 h (248.6 (101.3–331.2); *p* value > 0.05), 7 days (189.9 (73.9–299.5); *p* value > 0.05), 1 month (189.9 (79.8–225.4); *p* value > 0.05), 3 months (154.3.6 (86.6–193.9); *p* value > 0.05), and 6 months (132.8 (72.4–189.2); *p* value > 0.05). However, there was a significant decrease in the levels of NT-proBNP measured at 12 months (123.2 (59.0–171.5); *p* value < 0.05) and 24 months (100.9 (65.4–134.5); *p* value < 0.05), respectively (Table 3 and Table 4). For NT-proBNP, there were significant differences in repeated measurements (*p* value < 0.05). 

Adrenaline: The mean baseline adrenaline level measured before the procedure was 97.8 (52.8–120.0) pg/mL. Compared with baseline, there was no significant difference in adrenaline levels measured at 24 h (67.0 (58.0–103.4); *p* value > 0.05) and 7 days (66.7 (50.1–82.4); *p* value > 0.05). However, there was a significant decrease in adrenaline levels measured at 1 month (56.0 (45.2–67.9); *p* value < 0.05), 3 months (53.5 (47.9–64.4); *p* value < 0.05), 6 months (53.4 (41.9–63.2); *p* value < 0.05), 12 months (60.0 (52.7–70.2); *p* value < 0.05), and 24 months (52.0 (48.9–57.9); *p* value < 0.05) (Table 3 and Table 4). For adrenaline, there were significant differences in repeated measurements (*p* value < 0.05). 

Noradrenaline: The mean baseline noradrenaline level measured before the intervention was 584 (401–772) ng/mL. Compared with baseline, there was a significant decrease in noradrenaline levels, which was observed at 24 h (270 (94–398); *p* value < 0.05), 7 days (227 (163–548); *p* value < 0.05), 1 month (246 (188–518); *p* value < 0.05), 3 months (233 (188–232); *p* value < 0.05), 6 months (176 (150–208); *p* value < 0.05), 12 months (135 (97–253); *p* value < 0.05), and 24 months (115 (89–193); *p* value < 0.05) (Table 3 and Table 4). For noradrenaline, there were significant differences in repeated measurements (*p* value < 0.05).

### 3.4. Impact of LAA Closure on Adrenergic System and RAAS

Aldosterone: The mean baseline aldosterone level measured before the procedure was 165.6 (153.2–178.0) pg/mL. Compared with baseline, there was no significant difference in adrenaline levels measured at 24 h (173.7 (151.9–186.6); *p* value > 0.05), 7 days (163.9 (131.3–181.6); *p* value > 0.05), 1 month (158.6 (153.2–195.3); *p* value > 0.05), and 3 months (159.2 (158.1–185.5); *p* value > 0.05). However, there was a significant decrease in the levels of aldosterone measured at 6 months (146.0 (130.3–142.7); *p* value < 0.05), 12 months (126.3 (105.0–148.9); *p* value < 0.05), and 24 months (123.5 (114.9–130.2); *p* value < 0.05) (Table 3 and Table 4). For aldosterone, there were significant differences in repeated measurements (*p* value < 0.05). 

Plasma Renin Activity: The mean baseline value of plasma renin activity measured before the intervention was 1.63 (1.16–1.96) ng/mL/h. Compared to baseline, there was no significant difference in plasma renin activity values measured after 24 h (1.66 (1.25–1.99); *p* value > 0.05). However, there was a significant decrease in plasma renin activity measured at 7 days (1.29 (1.22–1.45); *p* value < 0.05), 1 month (1.30 (1.18–1.60); *p* value < 0.05), 3 months (1.19 (0.88–1.44); *p* value < 0.05), 6 months (1.25 (1.17–1.35); *p* value < 0.05), 12 months (1.12 (0.89–1.33); *p* value < 0.05), and 24 months (1.21 (1.02–1.33); *p* value < 0.05) (Table 3 and Table 4). For plasma renin activity, there were significant differences in repeated measurements (*p* value < 0.05). 

Vasopressin: The mean baseline vasopressin level measured before the intervention was 8.19 (6.26–13.86) pg/mL. Compared with baseline, there was a significant decrease in vasopressin levels measured at 24 h (6.97 (6.23–7.56); *p* value < 0.05), 7 days (3.44 (2.38–4.39); *p* value < 0.05), 1 month (3.05 (2.29–4.57); *p* value < 0.05), 3 months (4.26 (3.89–4.63); *p* value < 0.05), 6 months (2.87 (1.60–5.26); *p* value < 0.05), 12 months (2.85 (1.47–4.63); *p* value < 0.05), and 24 months (2.79 (2.03–4.18); *p* value < 0.05) (Table 3 and Table 4).

Endothelin: The mean baseline endothelin level measured before the intervention was 1.98 (1.31–2.52) ng/mL. Compared with baseline, there was a significant increase in endothelin levels at 24 h (2.21 (1.92–2.61); *p* value < 0.05), 7 days (2.35 (1.52–2.80); *p* value < 0.05), 1 month (2.56 (1.94–3.14); *p* value < 0.05), 3 months (2.78 (1.85–3.32); *p* value < 0.05) 6 months (2.84 (2.05–3.80); *p* value < 0.05), 12 months (2.94 (2.44–3.01); *p* value < 0.05), and 24 months (3.50 (2.89–4.05); *p* value < 0.05) (Table 3 and Table 4). For endothelin, there were significant differences in repeated measurements (*p* value < 0.05).

### 3.5. Impact of Epicardial LAA Closure on Systolic and Diastolic Blood Pressure

The mean systolic blood pressure (SBP) at baseline was 133.9 ± 20.6 mmHg. After LAAC, SBP was significantly lower at 3 months (118.2 ± 10.3 mmHg), 6 months (115.0 ± 11.4 mmHg), 12 months (118.3 ± 8.6 mmHg), and 24 months (117.9 ± 5.8 mmHg), respectively. Mean diastolic blood pressure (DBP) was 82.4 ± 11.0 mmHg at baseline. After LAAC, DBP was significantly lower at 3 months (68.9 ± 7.6 mmHg), 6 months (70.6 ± 8.8 mmHg), 12 months (70.4 ± 7.2 mmHg), and 24 months (75.6 ± 6.5 mmHg), respectively. Figure 1 illustrates the trend in blood pressure measurements at follow-up. 

## 4. Discussion

### 4.1. Main Findings

In patients with long-standing persistent AF, epicardial LAAC led to numerous acute and long-term neurohormonal and blood pressure changes. The epicardial LAAC procedure minimally affected electrolytes and acutely increased ANP, while, in the long term, there was a significant decrease in noradrenaline, adrenaline, and RAAS levels—while endothelin levels were increased during the follow-up period. There was a sustained decrease in systolic and diastolic blood pressure, reflecting the long-term decrease in adrenergic neurohormones and RAAS.

### 4.2. Impact of Epicardial LAAC on Serum Electrolytes

Launi et al. demonstrated that the sodium and potassium levels had not significantly changed in the 6 and 24 weeks following the WATCHMAN procedure, but no acute levels were measured after the procedure [5]. Maybrook et al. demonstrated a decrease in serum sodium levels immediately after the epicardial LAAC procedure that normalized at six months [6]. The current study also showed an acute decrease in serum sodium level within 24 h; whereas serum sodium level did not change significantly at 7 days, 1 month, 3 months, 6 months, 12 months, and 24 months. Although the acute changes in serum sodium level can be explained by a short-term increase in ANP level (leading to natriuresis), the absence of long-term changes in serum sodium level from baseline suggests the existence of alternative cardiac sites for natriuretic peptide secretion that may compensate for the loss of ANP secretion in the LAA [7]. Clinical factors that may influence the occurrence of hyponatremia after epicardial LAAC include low BMI (<25 kg/m^2^), increased LA diameter, and low systolic blood pressure [8]. 

### 4.3. Impact of Epicardial LAAC on Natriuretic Peptides and Adrenergic System

Previous investigators have studied the changes in natriuretic peptides, adrenergic system, and RAAS and have shown different results due to neurohumoral modulation, especially in epicardial LAAC compared to endocardial LAAC [7]. Endocardial LAAC also demonstrated different concentrations of natriuretic peptides during the follow-up period [5,9,10], possibly due to a different extent of leakage around the device and the absence of acute and chronic LAA necrosis. Mechanistically, epicardial LAAC is different from endocardial LAAC [11]. Epicardial LAAC leads to inflammation of the LAA, which is later followed by fibrosis, scarring, closure, and permanent resorption of the LAA over a prolonged period [12]. A previous study reported the changes in ANP three months after epicardial LAAC [13], but acute changes were not investigated. Suture application around the base of the LAA causes ischemia and pressure necrosis of the LAA, resulting in an initial massive release of ANP into the circulation within 24 h and 72 h, until stores are depleted. The results of this study confirmed the acute significant increase in ANP by 24 h, and further examined the mid- and long-term ANP changes, which showed a decrease in ANP from 7 days to 3 months before normalizing to baseline from 6 months to 24 months. Since fibrosis and resorption of the LAA occurs after epicardial LAAC, the normalisation of ANP is a compensatory production of ANP from tissues other than the LAA [14].

In our previous study, we demonstrated a reduction in adrenaline and noradrenaline levels at 24 h and 3 months after epicardial LAAC [2,6]. The results of this current study are consistent with those of our previous study and further demonstrate a sustained reduction in adrenaline and noradrenaline levels at long-term follow-up at 12 and 24 months. The sustained reduction in adrenaline and noradrenaline levels may be due to the depletion of the LAA of natriuretic peptides and destruction of the afferent fibres and ganglionic plexi surrounding the LAA due to epicardial LAAC, which results in the disruption of the natriuretic peptide pathway and neural reflexes [7]. Although the interaction between natriuretic peptides and adrenaline remains unknown, a previous clinical study has shown that epicardial LAAC can lead to downregulation of adrenaline through negative feedback [2]. 

### 4.4. Impact of Epicardial LAAC on Renin Angiotensin Aldosterone System

One of the important physiological functions of ANP is to inhibit angiotensin-II-induced drinking, pituitary hormone release, and fluid and sodium retention [15]. Given the antagonistic effect of ANP on the RAAS, any acute change in ANP levels in patients receiving epicardial LAAC should be accompanied by an opposite change in the RAAS that returns to normal over time. A sustained reduction in renin and aldosterone levels is observed in both the short term [2] and the long term (current study). As ANP levels normalise after six months, the RAAS levels remain reduced and the reduction in RAAS may be influenced by other factors such as sympathetic neurohormones, which are also reduced in the long term.

Fat pad necrosis around the LAA due to suture ligation, resulting in permanent damage to the afferent and efferent autonomic inputs, could contribute to the ongoing downregulation of the RAAS [2]. As the current evidence is insufficient to explain these anatomical abnormalities due to LAA suture ligation, it would be beneficial to conduct larger prospective studies to understand the complex interplay of these afferent and efferent innervations of the LAA ganglionic structures, baroreceptors, and RAAS and the resulting clinical outcomes [2]. 

Our study is unique in that there is no previous study that has investigated the changes in vasopressin levels after epicardial LAAC. According to Sun et al., blood serum endothelin-1 levels did not change significantly at 24 h, 3 months, 6 months, and 12 months after percutaneous epicardial LAAC [16], a finding that is at odds with our study. The exact mechanism for this consistent increase in endothelin-1 levels in this study is currently unknown. According to Mayyas, et al., higher endothelin-1 levels were associated with larger atrial size, fibrosis, hypertrophy, and persistence of AF [17], leading to increased all-cause and cardiovascular mortality [18].

In patients with systemic hypertension and heart failure, an upregulated RAAS can lead to AF [19]. Recent clinical trials have demonstrated that RAAS antagonism may reduce the incidence of new onset AF [20,21,22,23]. Therefore, it can be assumed that sustained downregulation of the RAAS is one of the supporting mechanisms of LAA suture closure devices to prevent the occurrence of a new onset AF. 

### 4.5. Impact of Epicardial LAAC on Systemic Blood Pressure

In a previous LAA hemostasis study by our group, we observed that blood pressure was lowered 24 h and 3 months after epicardial LAAC [2]. The acute reduction in blood pressure is due to the release of ANP and the development of hyponatremia, which generally resolves within 2–7 days [8]. In another prospective, non-randomised registry, a significant reduction in systolic blood pressure was observed 3 months and 1 year after the procedure in hypertensive patients who underwent epicardial LAAC compared with patients who received endocardial LAAC [24]. Similar to the above results, we found a significant reduction in systolic and diastolic blood pressure 1 year and 2 years after epicardial LAAC, independent of the serum levels of ANP and sodium. We hypothesise that the LARIAT procedure downregulates RAAS and adrenergic activity andthus is the possible underlying mechanism for the reduction in blood pressure over a long term period of time in our study. Further studies are needed to assess whether epicardial LAAC has long-term effects on blood pressure control that may translate into a reduction in antihypertensive medication, as no changes in medication occurred in the postoperative period in our study.

### 4.6. Study Limitations

A limitation of the study is the single-arm nature, using a method of epicardial LAAC with the LARIAT device. This does not allow for generalization to other methods of epicardial LAAC or comparison with endocardial LAAC procedures. The main objective of the study was to see how the RAAS system and hormone levels are changed during the follow-up period after LAA closure therapy. However, the lower systolic and diastolic blood pressure observed in this study at the 1- and 2-year follow-up period is only a consequence of lower RAAS and adrenergic activity. Nevertheless, our study provides a trend in neurohormonal biomarkers after epicardial LAA ligation with the LARIAT device that can be further investigated and explored in larger clinical trials that are already underway. 

Biomarker changes also did not correlate with clinical outcomes, hospital readmissions, mortality, and healthcare cost utilisation in patients with long-standing persistent AF who underwent percutaneous epicardial LAA ligation. Therefore, it is difficult to interpret whether changes in biomarkers have an impact on long-term clinical outcomes. Another limitation of the study is the lack of a homogeneous study group in terms of comorbidities. The simultaneous presence of chronic diseases such as diabetes, ischaemic diseases, or heart failure, as well as as the use of medication, may have an impact on hormone levels. In addition, the influence of confounding factors may be amplified by the small size of the study group. Future studies need to assess the long-term clinical and health economic aspects as well as the impact on the development of atrial myopathy. 

## 5. Conclusions

Epicardial LAAC in AF patients is associated with persistent neurohormonal changes that favor blood pressure reduction. Future prospective studies are needed to confirm the effects on blood pressure and to understand the mechanistic basis of these physiological changes observed in this study.

## Figures and Tables

**Figure 1 jcm-11-01495-f001:**
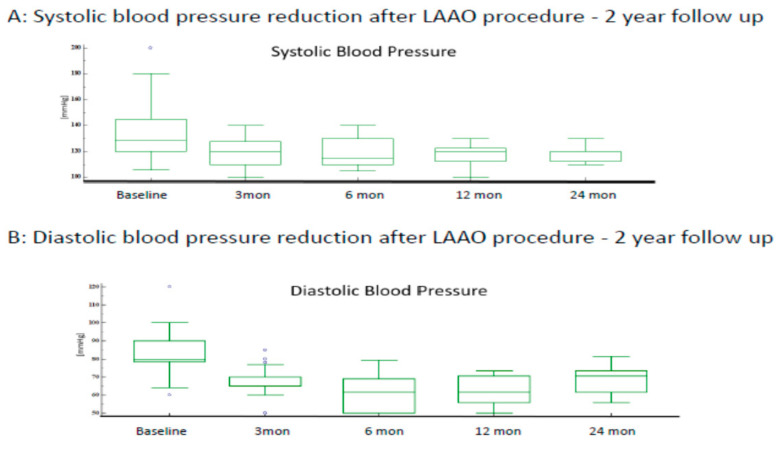
In a 2-year follow-up after LAAO procedure, mean SBP and DBP in our study is (119.4 ± 4.8 mmHg) and (73.2 ± 3.7 mmHg), respectively.

**Table 1 jcm-11-01495-t001:** Baseline characteristics of the cohort.

Variable	*n* = 60
Age, yrs	67.5 ± 8.1
Male, *n* (%)	40 (66.7)
Body mass index, kg/m^2^	28.7 ± 2.9
Coronary artery disease, *n* (%)	11 (18.3)
Hypertension, *n* (%)	48 (80)
Diabetes, *n* (%)	15 (25)
Congestive heart failure, *n* (%)	8 (13.3)
CHA_2_DS2-VASc	4.0 ± 1.7
HAS-BLED	3.6 ± 1.2
Long-standing persistent AF	60 (100)

**Table 2 jcm-11-01495-t002:** Sequential change in serum electrolytes after LAAC from baseline to 24 months follow up.

Variable	Baseline	24 h	7 Days	1 Month	3 Months	6 Months	12 Months	24 Months
Sodium, mmol/L	141.1 ± 2.6	138.4 ± 2.2 *	142.1 ± 3.2	142.1 ± 2.6	140 ± 4.1	139.5 ± 4.5	140.7 ± 2.6	140.7 ± 3.1
Potassium, mmol/L	4.3 ± 0.4	4.1 ± 0.7	4.4 ± 0.5	4.5 ± 0.4	4.4 ± 0.5	4.3 ± 0.5	4.4 ± 0.3	4.1 ± 0.4
Chlorides, mmol/L	107.6 ± 4.2	109.8 ± 2.9	107.9 ± 4.4	104.3 ± 3.5	104.2 ± 3.3	103.2 ± 2.9	103.7 ± 2.8	100.4 ± 4.2

* *p* < 0.05 compared to baseline.

**Table 3 jcm-11-01495-t003:** Sequential changes in natriuretic peptides and adrenergic markers from baseline to 24 months after LAAC. (*—the result was statistically significant).

Variable	Baseline	24 h	7 Days	1 Month	3 Months	6 Months	12 Months	24 Months
NT-proANP, ng/mL	9.94 (4.09–16.87)	15.39 (10.93–16.10) *	6.31 (3.71–11.79) *	5.44 (4.29–10.58) *	4.58 (4.00–5.48) *	8.36 (5.10–14.80)	9.03 (4.73–13.98)	11.01 (4.98–19.81)
NT-proBNP, pg/mL	167.8 (63.7–238.7)	248.6 (101.3–331.2)	189.9 (73.9–299.5)	189.9 (79.8–225.4)	154.3 (86.6–193.9)	132.8 (72.4–189.2)	123.2 (59.0–171.5) *	100.9 (65.4–134.5) *
Adrenaline, pg/mL	97.8 (52.8–120.0)	67.0 (58.0–103.4)	66.7 (50.1–82.4)	56.0 (45.2–67.9) *	53.5 (47.9–64.4) *	53.4 (41.9–63.2) *	60.0 (52.7–70.2) *	52.0 (48.9–57.9) *
Noradrenaline, ng/mL	584 (401–772)	270 (94–398) *	227 (163–548) *	246 (188–518) *	233 (188–282) *	176 (150–208) *	135 (97–253) *	115 (89–193) *
Aldosterone, pg/mL	165.6 (153.2–178.0)	173.7 (151.9–186.6)	163.9 (131.3–181.6)	158.6 (153.2–195.3)	159.2 (158.1–185.5)	146.0 (130.3–142.7) *	126.3 (105.0–148.9) *	123.5 (114.9–130.2) *
Plasma renin activity, ng/mL/h	1.63 (1.16–1.96)	1.66 (1.25–1.99)	1.29 (1.22–1.45) *	1.30 (1.18–1.60) *	1.19 (0.88–1.44) *	1.25 (1.17–1.35) *	1.12 (0.89–1.33) *	1.21 (1.02–1.33) *
Vasopressin, pg/mL	8.29 (6.26–13.86)	6.97 (6.23–7.56) *	3.44 (2.38–4.39) *	3.05 (2.29–4.57) *	4.26 (3.89–4.63) *	2.87 (1.60–5.26) *	2.85 (1.47–4.63) *	2.79 (2.03–4.18) *
Endothelin-1, pg/mL	1.98 (1.31–2.52)	2.21 (1.92–2.61) *	2.35 (1.52–2.80) *	2.56 (1.94–3.14) *	2.78 (1.85–3.32) *	2.84 (2.05–3.80) *	2.94 (2.44–3.01) *	3.50 (2.89–4.05) *

**Table 4 jcm-11-01495-t004:** Changes in electrolytes, natriuretic peptide, and adrenergic markers after LAAC compared to baseline.

Factor	24 h	7 Days	1 Month	3 Months	6 Months	12 Months	24 Months
NT-proANP	↑	↓	↓	↓	-	-	-
NT-proBNP	-	-	-	-	-	↓	↓
Adrenaline	-	-	↓	↓	↓	↓	↓
Noradrenaline	↓	↓	↓	↓	↓	↓	↓
Aldosterone	-	-	-	-	↓	↓	↓
Plasma renin activity	-	↓	↓	↓	↓	↓	↓
Vasopressin	↓	↓	↓	↓	↓	↓	↓
Endothelin	↑	↑	↑	↑	↑	↑	↑

All values compared to baseline levels, - no change, ↑ increased level, ↓ decreased level.

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
