# Peer review of "Long Term Impact of Epicardial Left Atrial Appendage Ligation on Systemic Hemostasis: LAA HOMEOSTASIS-2"

_jcm, 2022, doi:10.3390/jcm11061495_

Round 1

Reviewer 1 Report

I want to thank the authors for this article. I still have some requests that may help to improve this article.

  1. Improve this sentence grammatically : (tubes were anticoagulated with K3- EDTA for complete blood count, tubes with 0.109M sodium citrate for hemostasis and fibrinolysis tests, and serum vacuum tubes for routine clinical chemistry laboratory testing and ELISA tests)
  2. Some repetitions in the text, please consider improving them. Also, anti-factor Xa activity (IU/mL)was measured pre-procedure for all patients. All patients were instructed to maintain their routine diets and not to take any additional salt or carbohydrate loading up to 48 hours before the blood draw
  3. Define this sentence better as it's really relevant to your results and interpretations. No significant medication changes were noted in the follow up.
  4. Because many of the studied elements may be affected by different drugs, did you consider this as a study limitation? A small part of the patients suffered from congestive heart disease, this is also as knows greatly related to enormous neurohormonal dysregulation affecting many parameters you studied in this work, how would you comment on this? 

Author Response

Dear Reviewers,

We thank the reviewers for their insightful comments and review. We agree with their comments and have amended the manuscript accordingly. Please find responses (in bold and italics) after each comment.

Please also note, that the current version of the article has also undergone professional linguistic correction.

  1. Improve this sentence grammatically : (tubes were anticoagulated with K3- EDTA for complete blood count, tubes with 0.109M sodium citrate for hemostasis and fibrinolysis tests, and serum vacuum tubes for routine clinical chemistry laboratory testing and ELISA tests)

Thank you for that comment. We corrected this sentence grammatically:

Blood samples were collected in vacutainer tubes (tubes were anticoagulated with K3-EDTA for complete blood count, 0.109M sodium citrate was added for haemostasis and fibrinolysis tests, and serum vacuum tubes were used for routine clinical chemistry laboratory tests and ELISA test.

  1. Some repetitions in the text, please consider improving them. Also, anti-factor Xa activity (IU/mL)was measured pre-procedure for all patients. All patients were instructed to maintain their routine diets and not to take any additional salt or carbohydrate loading up to 48 hours before the blood draw

Thanks for your comment. The sentence has been corrected.

In addition, anti-factor Xa activity (IU/mL) was measured in all patients before the procedure. All patients were instructed to maintain their usual diet and not to consume any additional salts or carbohydrates for up to 48 hours before the blood draw.

  1. Define this sentence better as it's really relevant to your results and interpretations. No significant medication changes were noted in the follow up.

We thank the reviewer for this comment. We agree that this sentence is extremely important to the results we obtain. We have now corrected this expression to be more understandable and more precise.

Patients were on stable medical BP reduction therapy prior to the procedure and were instructed not to change their medications throughout the follow-up period. All blood pressure medications were routinely continued, with the exception of diuretics, which were discontinued on the day of the procedure. Patient adherence to medication was strictly monitored by interview and patient diary. No changes in blood pressure medication were noted during follow-up that could affect the obtained results.

  1. Because many of the studied elements may be affected by different drugs, did you consider this as a study limitation? A small part of the patients suffered from congestive heart disease, this is also as knows greatly related to enormous neurohormonal dysregulation affecting many parameters you studied in this work, how would you comment on this? 

Thank you for this comment. The reviewer's comment is very valuable and correct. The above comments are one of the most important limitations of our study. We have added our comments to the section: Limitations.

Another limitation of the study is the lack of a homogeneous study group in terms of comorbidities. The simultaneous presence of chronic diseases such as diabetes, ischaemic diseases or heart failure, as well as medication, may have an impact on hormone levels. In addition, the influence of confounding factors may be amplified by the small size of the study group.

Reviewer 2 Report

Dr. Bartus and coworkers investigate the impact of percutaneous epicardial LAA exclusion on the neurohormonal profiles on long-term follow-up.

I have some suggestions that, in the opinion of this reviewer, will improve the paper.

Ask your statistician to perform a longitudinal analysis, more appropriate to follow the changes of the hormone etc. over time.

Put in table 4 the number and in the test the statistical significance and the differences. Arrows make the message difficult to read

Differentiate between persistent and long-standing persistent.

Please define acronyms at their first appearance.

Author Response

Dear Reviewers,

We thank the reviewers for their insightful comments and review. We agree with their comments and have amended the manuscript accordingly. Please find responses (in bold and italics) after each comment.

Please also note, that the current version of the article has also undergone professional linguistic correction.

Dr. Bartus and coworkers investigate the impact of percutaneous epicardial LAA exclusion on the neurohormonal profiles on long-term follow-up.

I have some suggestions that, in the opinion of this reviewer, will improve the paper.

Ask your statistician to perform a longitudinal analysis, more appropriate to follow the changes of the hormone etc. over time.

Thank you for that comment. We strongly agree with reviewer comments. We performed longitudinal analysis for all variables. We add this information in statistic section and results sections.

Put in table 4 the number and in the test the statistical significance and the differences. Arrows make the message difficult to read.

Thank you for that comment. We respect the reviewer's comments but feel that presenting the results in the form of arrows in the table is more readable and shows change than the form of numbers and statistical results that the reader finds in the text and in Table 3.
We have previous experience of presenting this type of work at various conferences and in publications in the Journal of the American University of Cardiology.
We believe that the graphical presentation also increases the chances of the number of citations of the published work.

Our earlier work was cited more than 54 times (Lakkireddy, D., Turagam, M., Afzal, M. R., Rajasingh, J., Atkins, D., Dawn, B., ... & Holmes, D. J. (2018). Left atrial appendage closure and systemic homeostasis: the LAA HOMEOSTASIS study. Journal of the American College of Cardiology, 71 (2), 135-144. ) and 27 times (Bartus, K., et al. "Atrial natriuretic peptide and brain natriuretic peptide changes after epicardial percutaneous left atrial appendage suture ligation using LARIAT device." J Physiol Pharmacol 68.1 (2017): 117-23.)

Differentiate between persistent and long-standing persistent.

Thank you for that comment. We apologise for it. All the cases were long-standing persistent AF. We have corrected this in the manuscript.

Please define acronyms at their first appearance.

Thank you for that comment. We apologise for it. We define all acronyms at their first appearance according to the reviewer's suggestion.
